# MODEL ZOOS FOR BENCHMARKING PHASE TRANSITIONS IN NEURAL NETWORKS

## ABSTRACT

Understanding the complex dynamics of neural network training remains a central challenge in deep learning research. Work rooted in statistical physics has identified phases and phase transitions in neural network (NN) models, where models within the same phase exhibit similar characteristics but qualitatively differ across phases. A prominent example is the double-descent phenomenon. Recognizing these transitions is essential for building a deeper understanding of model behavior and the underlying mechanics. So far, these phases are typically studied in isolation or in specific applications. In this paper, we show that phase transitions are a widespread phenomenon. However, identifying phase transitions across different methods requires populations that cover different phases. For that reason, we introduce Phase Transition Model Zoos, a structured collection of neural networks trained on diverse datasets and architectures. These model zoos are carefully designed to help researchers systematically identify and study phase transitions in their methods. We demonstrate the relevance of phase transitions across multiple applications, including fine-tuning, transfer learning, out-of-distribution generalization, pruning, ensembling, and weight averaging. The diversity of applications underscores the universal nature of phase transitions and their impact on different tasks. By providing the first structured dataset specifically designed to capture phase transitions in NNs, we offer a valuable tool for the community to systematically evaluate machine learning methods and improve their understanding of phase behavior across a wide range of applications and architectures.

## 1 INTRODUCTION

Neural network (NN) research has made considerable progress in recent years. To continue making sustainable progress in NN research, there is a need for thorough understanding of methods. Currently, proposed methods are usually evaluated on a few datasets, or on leaderboards with a few models. While it has become standard practice to report performance averaged over several random seeds, these single performance metrics lead to very sparse feedback in a very complex hypothesis and methods space and ultimately a gap in understanding. Addressing this challenge requires systematic evaluation that strategically covers the hyperparameter and model space, demonstrating where a method succeeds or fails in different regimes.

**Identifying Phase Transitions in Neural Networks** To identify relevant regimes and necessary hyperparameter variations for method evaluation, phase transitions in neural networks provide a useful perspective. Phase transitions have been studied extensively in the machine learning literature (Schwarze et al., 1992; Seung et al., 1992; Martin & Mahoney, 2019a;b). Within phases, models are relatively homogeneous, with abrupt changes from one phase to the next. One example for such a phase transition is the *double descent* (Nakkiran et al., 2019), which describes the transition from high to low generalization error with increasing model capacity. Other work describes the transitions between where different NN methods perform well or fail: in training (Zhou et al., 2024), ensembling (Theisen et al., 2023), pruning (Zhou et al., 2023), etc. Notably, while phases exist in performance metrics, loss landscape metrics, as well as downstream applications, they do not necessarily overlap. For instance, ensembling reduces the sensitivity to training parameters, while pruning benefits from noisier pretraining. Therefore, identifying the phases and phase transitions in methods and how different methods affect them provides a more robust evaluation signal than individual data points that may lie in any of the phases.

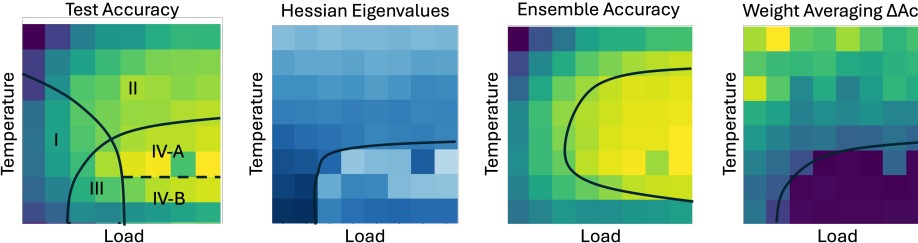

Figure 1: Varying hyperparameters load and temperature parameters reveal phases and phase transitions in neural networks, as introduced in Section 2. These phases exist in the outcome of training (left figure) , describing the transition from low to high accuracy. Phases can also be identified in loss-landscape metrics, like the eigenvalues of the hessian to estimate curvature (second from left). Methods applied on pre-trained models, like ensembling (second from right) or weight averaging (right) likewise contain phase transitions as explored in Section 4. To systematically evaluate neural network methods, evaluating in different phases and localizing phase transitions is necessary.

**Testing Methods on Populations of Neural Networks**   Evaluating methods for phase transitions does present its own challenges, however. In particular, many machine learning methods depend on pre-trained models. The phase transitions of these methods depend on the phase transitions of the pre-trained models. The exploration of the phase distribution in these methods requires systematic evaluation over different pre-trained models - model populations. However, the quality of the feedback and scope of phase exploration depends on the diversity of the population. Most freely accessible models are part of large public repositories like Hugging Face (Wolf et al., 2020) or the PyTorch model hub (Pytorch). Within those collections, however, models are of varying quality and mostly unstructured, making systematic evaluation a challenge. Structured populations have been published as model zoo datasets (Schürholt et al., 2022b; Croce et al., 2020; Ouyang et al., 2022; Honegger et al., 2023). These studies, however, consider the diversity of the model in their populations only in terms of their generating factors. Phase transitions are not explicit target for these datasets and therefore only occur incidentally. To systematically evaluate methods for phase transitions in pre-trained models, there is therefore a need for populations with systematic variations that covers a broad range of phases.

**Contributions**   As a step towards robust evaluation, we present our Phase Transition Model Zoos dataset. The dataset contains populations of trained models, containing ResNet and ViT architectures of varying sizes, trained on SVHN, CIFAR10, CIFAR100 or TinyImagenet (Netzer et al., 2011; Krizhevsky & Hinton, 2009; Le & Yang, 2015). For each dataset-architecture combination, we have carefully trained grids of models with variations s.t. they contain the known phase transitions. We validate the phase transition with known descriptive loss landscape metrics and annotate the models with them. The model zoos contain a total of 1829 models between 11K and 360M parameters. To the best of our knowledge, this is the largest structured dataset of models.

Furthermore, we demonstrate the importance of phase transitions as well as the usefulness of our datasets to identify them. We evaluate several fundamental neural network methods on them and show that phase transitions appear in fine-tuning, transfer-learning, pruning, ensembling and weight averaging, some of which have to the best of our knowledge not been previously documented. We provide an overview of the results in Figure 1. Ensembling, for instance, is known to improve robustness and decrease noisiness, which manifests in a larger high performance phase compared to pre-trained models. Other methods show phases that distinctly overlap loss landscape metrics. Weight averaging can be expected to improve performance in one regime towards the top of the grid, and not improve performance in the other. This example shows that since different phases behave qualitatively differently, systematic method evaluation should cover these different regimes and identify the phase transitions.

The Phase Transition Model Zoos repesents a large collection of models, with the potential of being highly valuable to methods that leverage diverse and structured populations of NNs. In particular, as it is systematically generated to cover the different phases, it can help researchers systematically identify phase transitions in their methods, and comprehensively benchmark those. We hope this will allow for better understanding of the layout of phases in methods, how different methods affect the phase distribution, what the underlying mechanics for it are, and how to improve over them.

## 2 LOSS LANDSCAPE TAXONOMY

**Phases in Neural Networks loss landscapes**   The motivation for introducing phases and phase transitions in NN loss landscapes is rooted in statistical mechanics, where such phenomena explain qualitative changes in system behavior (Martin & Mahoney, 2019a).  Phases represent distinct regions in the parameter space where the system's properties are homogeneous or change smoothly, while phase transitions mark abrupt changes in these properties. In NNs, phases manifest in terms of generalization performance.  A prominent example for such a phase transition is the double descent pheonmenon (Nakkiran et al., 2019), a phase transition along the load axis (Liao et al., 2020; Derezinski et al., 2020). Similar empirical observations have been made recently on the *emergent* behaviors of large language models (Wei et al., 2022), in which non-smooth transitions can occur when some training hyperparameters (such as the model size) are modified.  However, it is not conclusive whether these emergent behaviors are indeed sharp phase transitions or merely due to specific ways of experimental measurements (Schaeffer et al., 2024). These phases and transitions are expected due to the complex, high-dimensional nature of NN optimization, where varying control parameters like data noise and training iterations can lead to qualitatively different behaviors, akin to physical systems undergoing phase changes. Motivated by statistical physics, Martin & Mahoney (2019a) identify two main types of hyperparameters in NN training: the noisiness of the training process, dubbed temperature, and the amount of data relative to the size of the model, dubbed relative load.  Using that notion, distinct phases with qualitatively different model properties on the temperature-load landscape can be identified (Yang et al., 2021). Interestingly, the phases and phase transitions can be linked to the structure of the loss landscape (Yang et al., 2021). Specifically, metrics such as the training loss, the sharpness of local minima, and mode connectivity or prediction similarity computed on the training data can be used to identify the phase of a model. This, in turn, allows for inference of model quality and the design of training algorithms that adapt when the phases change (Zhou et al., 2023).

**Loss landscape metrics**   Yang et al. (2021) categorize phases in load-temperature variations using four metrics. The first metric is the training loss, which evaluates whether the training data is interpolated. The other metrics describe the sharpness of the local minima, the similarity between models trained using different random seeds, and the connectivity between different local minima of the loss landscape. It should be noted that Yang et al. (2021) used a certain set of metrics to measure these loss landscape properties, but there are alternative metrics available. For example, the sharpness of local minima can be measured using *adaptive sharpness metrics* (Andriushchenko et al., 2023; Kwon et al., 2021), while similarity can be measured using *disagreement* (Theisen et al., 2023).

We define the loss landscape metrics following Yang et al. (2021). Let $\boldsymbol{\theta} \in \mathbb{R}^m$ denote the learnable weight parameter, $\mathcal{L}$ be the loss function. We compute metrics using the train set unless stated otherwise.

**Hessian-based metrics**  The Hessian matrix $\mathbf{H}$ at a given point $\boldsymbol{\theta}$ can be represented as $\nabla_{\boldsymbol{\theta}}^2 \mathcal{L}(\boldsymbol{\theta}) \in \mathbb{R}^{m \times m}$. The largest eigenvalue $\lambda_{\max}(\mathbf{H})$ and trace $\mathrm{Tr}(\mathbf{H})$ are used to summarize the local curvature properties in a single value. Specifically, a larger value of the top eigenvalue or trace indicates greater sharpness.

**Mode connectivity**  The mode connectivity assesses the presence of low-loss paths between different local minima and reflects how well different solutions are connected in the parameter space, indicating smoother and more generalizable loss landscapes. It is common to fit Bézier curves $\big(\gamma_\phi(t)$ — piecewise linear curves with trainable nodes — between two models $\boldsymbol{\theta}$ and $\boldsymbol{\theta}'$, and subsequently compute mode connectivity mc as

$$\mathrm{mc}(\boldsymbol{\theta}, \boldsymbol{\theta}') = \frac{1}{2}(\mathcal{L}(\boldsymbol{\theta}) + \mathcal{L}(\boldsymbol{\theta}')) - \mathcal{L}(\gamma_\phi(t^*)),$$

where $t^* = \underset{t}{\mathrm{argmin}} \left| \frac{1}{2}(\mathcal{L}(\boldsymbol{\theta}) + \mathcal{L}(\boldsymbol{\theta}')) - \mathcal{L}(\gamma_\phi(t)) \right|$. Here, mc $< 0$ indicates a loss barrier between the two models and hence poor connectivity. mc $> 0$ reveals lower loss regions between the models which indicates poor training. mc $\approx 0$ indicates well-connected models.

**CKA similarity**  Centered Kernel Alignment (CKA) (Kornblith et al., 2019a) is used to evaluate the similarity between representations learned by different NNs, providing a measure of consistency and robustness in feature learning. CKA helps to understand how similar the learned representations are

across different minima, linking representation similarity to landscape structure and generalization performance. The CKA between output logits $\mathbf{X}$ and $\mathbf{Y}$ generated by $\boldsymbol{\theta}$ and $\boldsymbol{\theta}'$ is computed as

$$\mathrm{cka} = \frac{\mathrm{HSIC}(\mathbf{K}, \mathbf{L})}{\sqrt{\mathrm{HSIC}(\mathbf{K}, \mathbf{K}) \cdot \mathrm{HSIC}(\mathbf{L}, \mathbf{L})}}$$

where HSIC is the Hilbert-Schmidt Independence Criterion and $\mathbf{K}$ and $\mathbf{L}$ are the Gram matrices of $\mathbf{X}$ and $\mathbf{Y}$, respectively.

**Phase taxonomy** Based on loss landscape metrics, the NN hyperparameter space is divided into five distinct phases, as depicted in Figure 2.

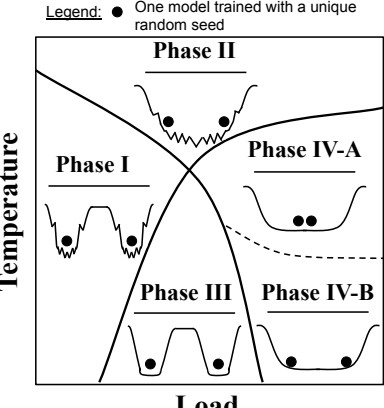

- **Phase I (underfitted & undersized)**: Train loss is high; Hessian metrics are relatively large (indicated by a rugged basin); Mode connectivity is negative (indicated by a barrier between two local minima).
- **Phase II (underfitted)**: Train loss is high; Hessian metrics are relatively large; Mode connectivity is positive.
- **Phase III (undersized)**: Train loss is small; Hessian metrics are relatively small (smooth basin); Mode connectivity is negative.
- **Phase IV-A (generalizing)**: Train loss is small; Hessian metrics are relatively small; Mode connectivity is near-zero (no barrier between minima); CKA similarity is relatively large.
- **Phase IV-B (overfitted)**: Train loss is small; Hessian metrics are relatively small; Mode connectivity is near-zero; CKA similarity is relatively small.

Figure 2: Five-phase taxonomy in NN hyperparameter space (Yang et al., 2021), varied by load-like and temperature-like parameters. Our zoos cover all five phases.

## 3 PHASE TRANSITION MODEL ZOOS

To create a population of models that covers relevant phases and can be used to evaluate for phase transitions, we train strucured populations of NNs with several architectures on different datasets following the blueprint introduced by Unterthiner et al. (2020). Within each *model zoo* population, we systematically vary load-like and temperature-like hyperparameters to realize all of the phases. For every model in the zoo, our dataset includes multiple checkpoints (i.e. saved model weights), at different training epochs. We annotate these samples with performance metrics (training and test loss and accuracy), as well as the loss landscape metrics outlined in Section 2. We further track loss and accuracy on train, test, and — if available — validation sets. In the following, we first detail our model zoo generation scheme. Subsequently, we analyze our models with conventional performance metrics, but also with loss landscape metrics to quantify the qualitative diversity of our zoos and validate that all of the phases are realized.

### 3.1 MODEL ZOOS GENERATION

We create 10 zoos from combinations between two architectures {ResNet, ViT} of different sizes and four standard computer vision datasets {SVHN, CIFAR-10, CIFAR-100, TinyImagenet}. Details on the model zoos configurations can be found in Table 1 in the Appendix. We choose ResNet (He et al., 2016) and ViT (Dosovitskiy et al., 2021) architectures for the zoos because of their proliferation in computer vision to achieve representative populations. Importantly, ResNet and ViT architectures allow smooth scaling of model width and thus model capacity for the same architecture without the need to adjust the learning scheme. The selection of datasets follows the same logic. We emphasize that our model zoo blueprint is not limited to either specific architectures nor to computer vision tasks.

To obtain models in all phases, we introduce specific variations in the training hyperparameters. Previous work identifies the phases on the surface spanned by load-like and temperature-like hyperparameters (Martin & Mahoney, 2019a; Yang et al., 2021) The load-like parameters can be understood as the amount training data relative to the model capacity. Temperature represents the noisiness of

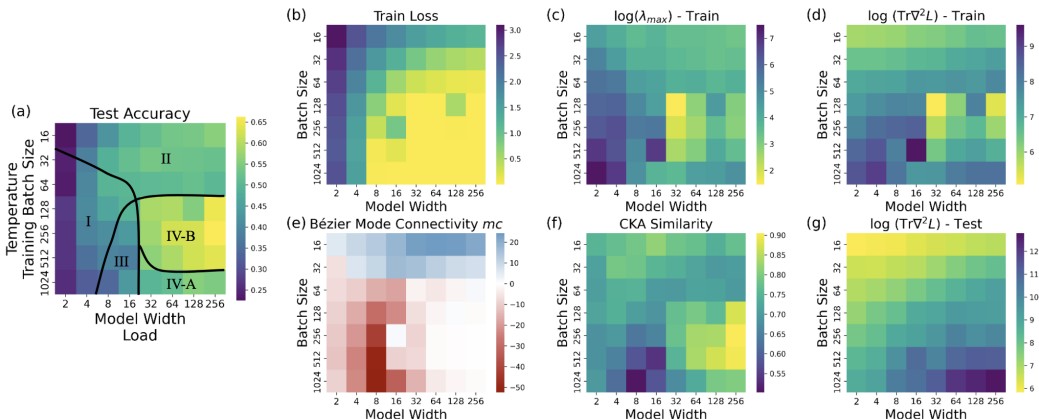

Figure 3: Performance and loss landscape metrics for the CIFAR-100 ResNet-18 model zoo. **(a):** test accuracy and phases of models in the zoo. **(b):** training loss; **(c-g)** different loss landscape metrics introduced in Section 2. Our model zoos cover all phases identified in previous work (Yang et al., 2021).

the training process. Following previous work, we realize variations in load by changing the model width. Increasing the model width increases model capacity and thus decreases the relative load. By varying the width, we achieve variations in model capacity without changing the architecture or having to adapt the training scheme. In ResNets, the width directly refers to the number of channels. In ViTs, we realize width by changing the `model_dim` parameter, i.e. the size of intermediate representations. To realize variations in temperature, we choose to adapt the batch-size. Here, lower batch-size increases the noisiness of the training updates and this increases temperature. For every combination on the grid, we train three different models using random seeds $\{1, 2, 3\}$. All other hyperparameters are kept constant between the models.

## 3.2 PHASES SYSTEMATICALLY EMERGE IN EVERY ZOO

The model zoos are designed to cover different phases. In the following, we validate phase coverage by testing for the phases introduced by Yang et al. (2021) summarized in Section 2. Full phase plots for all 10 zoos and further details can be found in Appendix A.3. Our experiments demonstrate that phase transitions are consistently present in the training of neural networks across all models and datasets evaluated, with the exact phase layout affected by the architecture, dataset, and data augmentation strategies. The specific characteristics of these phases remain consistent with the four-phase taxonomy outlined in previous studies, validating our experiment setup.

As illustrated in Figure 3 and in Figures 9-18, the phases manifest clearly in the combination of loss landscape metrics such as Hessian trace, mode connectivity, and CKA similarity. In particular, Phase IV-B, associated with the best test accuracy, is marked by low loss and high generalization performance. On the ResNet zoos, our results reveal that learning rate decay plays a significant role in shaping the phase distribution. Specifically, decaying the learning rate by $1e4$ under cosine annealing increases the area of Phase IV (well-trained regime) while reducing the presence of Phase II (under-trained regime), as the effect of batch size variations diminishes. This may be an indication for why learning rate decay is so successful. Our experiments show, that the phase transitions generalize across different datasets, architectures and training regimes. ViTs trained without strong data augmentation show an interesting additional sharp transition from phase II to phase IV, see Figure 11. Adding strong data augmentation appears to smoothen that transition again, but affects mode connectivity and sharpness, see Figure 14.

The presence of the different phases and transitions across all combinations of architectures and datasets studied validates the use of load-like and temperature-like hyperparameters for a systematic evaluation of the training dynamics. By incorporating awareness of phase transitions, training strategies can be better optimized for generalization and robustness, offering a valuable tool for improving both research methodologies and practical machine learning applications.

## 4  PHASE TRANSITIONS IN NEURAL NETWORK METHODS

In this paper, we argue that phase transitions are a widespread phenomenon across a variety of NN methods. In this section, we put that position to the test and illustrate the relevance of phase transitions across multiple applications.

In the following, we demonstrate phase transitions in several fundamental ML methods, to demonstrate (i) the existence of these phase transitions in these methods, and (ii) validate that they can be identified using our phase transition model zoos. The diversity of applications underscores the universal nature of phase transitions and their impact on different methods.

### 4.1  FINE TUNING

Fine-tuning pre-trained models is a widely used technique for improving model performance on new tasks or adjusting to distribution shifts(Yosinski et al., 2014). However, fine-tuning is also sensitive to the initial state of the pre-trained model, and the phase of the pre-trained model can significantly influence the final performance after fine-tuning.

In this context, phases can be expected to manifest in two ways: (i) in the fine-tuning configuration itself, where varying hyperparameters like learning rate or batch size affect the training dynamics, and (ii) in the phase of the pre-trained model, which directly impacts the performance of the fine-tuned model. These two sources of phase behavior are critical, as fine-tuning is essentially a continuation of training from a specific initialization.

Given the prevalence of fine-tuning from pre-trained models taken from model hubs, understanding the phase of the pre-trained model is essential for predicting how well a model will adapt to the target task. To investigate this, we fine-tuned models from two pre-trained zoos (trained on CIFAR-10) onto STL-10. We kept the fine-tuning setup constant to isolate the effects of the phase of the pre-trained model on the fine-tuning outcome.

Our results show that the fine-tuned models exhibit clear phase transitions in their performance, and these phases overlap substantially with the phases of the pre-trained models, see Figures 4, 9 and 11. This highlights that the phase of the pre-trained model plays a crucial role in determining the phase and, consequently, the performance of the fine-tuned model. For instance, models that were pre-trained in Phase IV (well-trained regime) continued to exhibit strong generalization when fine-tuned, while models from under-trained phases (e.g., Phase II) struggled to adapt, leading to poorer fine-tuning performance.

While our fine-tuning setup was kept intentionally simple, more sophisticated fine-tuning methods could likely shift the phase distribution. Evaluating these methods for phase transitions, as enabled by our model zoos, would provide valuable insights into the impact of different fine-tuning strategies.

### 4.2  TRANSFER LEARNING

While fine-tuning is commonly used for specialization in the same task, transfer learning is a powerful approach for adapting pre-trained models to new tasks or datasets. That way, it allows models trained on general tasks to be transferred to more specific or domain-specific applications (Yosinski et al., 2014; Kornblith

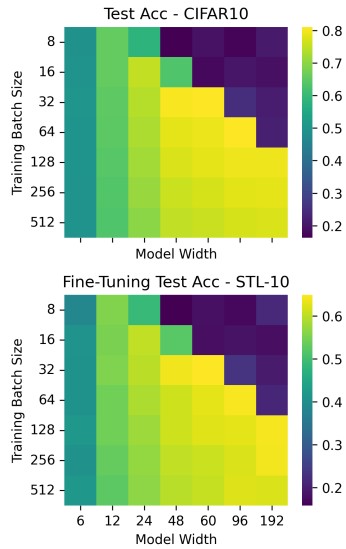

Figure 4: VIT - CIFAR-10 zoo phase plot. Top: Test accuracy on CIFAR10. Bottom: Test accuracy on STL-10.

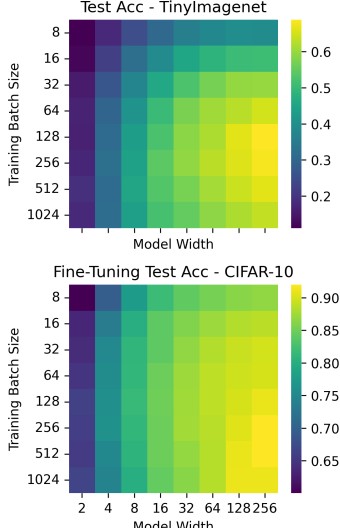

Figure 5: ResNet18 - Tiny-Imagenet zoo phase plot. Top: Test accuracy on TI. Bottom: Test accuracy on STL-10.

et al., 2019b; Recht et al., 2019). This scheme has proven highly successful, particularly in scenarios where labeled data is limited. However, as with fine-tuning, the success of transfer learning depends on the training configuration but also on the phase of the pre-trained model.

To identify the impact of the pre-trained model, we focus on how the pre-trained model's phase influences the outcome. To explore this, we transfer two model populations trained on TinyImageNet to CIFAR-10 and STL-10, keeping the transfer configuration constant to isolate the effects of pre-training phase.

Our results show distinct phase transitions in transfer learning, though the phases do not fully overlap with those of the pre-trained models, see Figures 5, 15 and 16. Choosing the best pre-trained model does not result in the best transfer-learned model. Indeed, lower pretraining temperature seems to help transfer performance. Task alignment and complexity may influence this, with closer tasks suggesting greater phase overlap, while more distant or complex tasks would favor models in lower-temperature phases.

This demonstrates that phase transitions are present in transfer learning but are more complex than in fine-tuning. Benchmarking transfer learning strategies using phase-aware model zoos provides deeper insight into how pre-trained models adapt to diverse tasks, guiding more effective transfer practices.

### 4.3 PRUNING

Model pruning is a common strategy to reduce the size of trained models, making them more efficient for deployment in resource-constrained environments such as mobile devices and edge computing (Lecun et al., 1989; Han et al., 2015; Frankle & Carbin, 2019; Molchanov et al., 2017). Typically, pruning has a noticeable impact on performance only at high pruning ratios, where a significant proportion of weights are removed.

Since pruning operates on pre-trained models, we find that the phase of the pre-trained model can significantly influence the pruning outcome. We conduct uniform magnitude pruning on the CIFAR-10 ResNet18 model zoo, removing 80% of the weights per layer and evaluating the pruned models, see Figure 6. As expected, the results reveal clear phase transitions. Larger models exhibit a higher capacity for pruning while maintaining performance, but importantly, the temperature of the pre-trained model plays a crucial role. Models pre-trained in higher-temperature phases (Phase II and IV-A) tend to perform better post-pruning compared to those from lower-temperature phases. The phase suggests a connection to not only model size, but also mode connectivity.

This demonstrates that selecting the best model before pruning does not necessarily result in the best post-pruning outcome. The complex phase transitions observed suggest that phase-aware pruning could lead to more effective strategies for optimizing models in deployment. Evaluating phase transitions in pruning offers a valuable tool for understanding how pre-trained model properties interact with pruning methods.

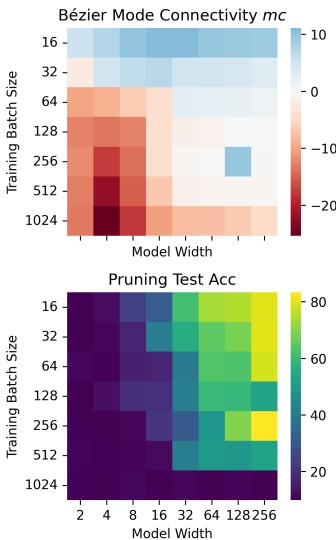

Figure 6: ResNet18 - CIFAR10 zoo phase plot. Top: Test accuracy of individual models. Bottom: Test accuracy of models after pruning with 80% pruning ratio.

### 4.4 ENSEMBLING

Ensembling models by averaging model predictions is well known to improve robustness in various tasks (Hansen & Salamon, 1990). However, the success of ensembling depends on the models used, suggesting that their phases are crucial for effective ensemble performance.

Previous work (Theisen et al., 2023) has shown that phase transitions play a significant role in ensembling, outlining conditions under which ensembling leads to improvements. To evaluate this, we replicate these findings using models from the Resnet18 zoo trained on CIFAR10, averaging models with the same temperature-load combination across three random seeds.

Our results reveal distinct phase transitions, confirming that ensembles composed of models from specific phases exhibit better robustness and generalization, see Figure 7 and 11. In particular, models from higher-temperature phases tend to produce more robust ensembles, while those from lower phases may lead to diminished performance gains.

This highlights the importance of considering model phases when constructing ensembles. Benchmarking ensembles using phase-aware model zoos provides a systematic approach to understanding how phase interactions affect ensemble outcomes, guiding more effective ensemble strategies.

Figure 7: ResNet18 - CIFAR10 zoo phase plot. Top: Test accuracy of individual models. Bottom: Test accuracy of ensembles.

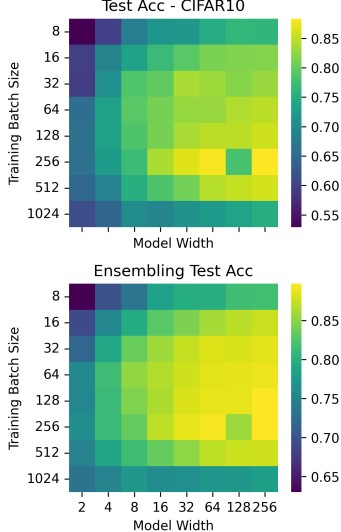

### 4.5 WEIGHT AVERAGING

While ensembling combines predictions, weight averaging directly combines model weights, offering benefits such as improved in- and out-of-domain performance without increasing inference costs (Wortsman et al., 2022b; Guo et al., 2023). However, weight averaging is sensitive to the alignment of model parameters, and inconsistent results have been observed depending on model initialization and optimization choices (Ainsworth et al., 2022). Understanding the phases of models being averaged is particularly important for ensuring robustness and reproducibility in these methods.

Previous work has demonstrated that the success of weight averaging is influenced by the structure of the loss landscape, including metrics like mode connectivity and Hessian trace (Wortsman et al., 2022a). To investigate this further, we apply weight averaging to models from our CIFAR-10 ResNet-18 zoo, examining how phases affect the success of averaging. Specifically, we average weights in two ways: **(i)** across 5 epochs within the same model (Wortsman et al., 2022b), **(ii)** across models within the same temperature-load cell, which we align using Git Re-Basin (Ainsworth et al., 2022).

Our results reveal distinct phase transitions that correlate with loss landscape metrics (see Figure 8, 9-17). Averaging over epochs improves performance in early phases (I and II), where the Hessian trace is large, while averaging across seeds using Git Re-Basin alignment negatively impacts performance in Phases I and III, where mode connectivity is poor.

Figure 8: ResNet18 - CIFAR10 zoo phase plots with performance and loss landscape metrics for weight averaging.

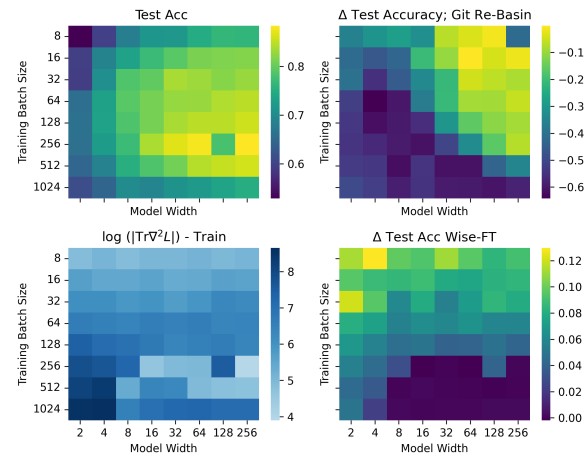

These findings underscore the importance of considering the phase of models in weight averaging to ensure reproducibility and robustness. The phase-dependent performance variations suggest that aligning models purely based on initialization may not be sufficient. Our dataset, with its detailed phase annotations, provides a valuable resource for further investigation into how phases influence the success of weight averaging, enabling more robust and generalizable model combination strategies.

## 5 APPLICATIONS FOR POPULATION BASED METHODS

Our dataset offers diverse opportunities for research beyond evaluation and the identification of phase transitions, particularly for methods that leverage populations of models and could benefit from annotated phase and loss landscape metrics. Here, we outline some of these applications.

### 5.1 POPULATION-BASED TRAINING

Population-based methods for hyperparameter tuning have shown great promise (Jaderberg et al., 2017; Li et al., 2020), yet they typically rely solely on validation performance for guidance. Our dataset provides loss landscape metrics, such as Hessian trace and mode connectivity, which could be used to guide models toward optimal phases during training. Recent works have demonstrated that leveraging such metrics can improve hyperparameter tuning efficiency and training outcomes (Zhou et al., 2023; Yao et al., 2018). Our model zoos, with multiple checkpoints and variations, offer a rich resource to explore these dynamics and further optimize population-based methods.

### 5.2 MODEL ANALYSIS

Predicting model properties based on weight statistics, without relying on test data, is an emerging field of research (Eilertsen et al., 2020; Unterthiner et al., 2020). Our dataset, which includes detailed annotations of loss landscape metrics across different phases, is ideal for training robust predictors that generalize across diverse model populations. This could lead to improved methods for predicting generalization power (Jiang et al., 2020) or detecting adversarial backdoors (Langosco et al., 2023). Additionally, approximating expensive metrics such as mode connectivity through weight-based predictors could reduce the computational cost of model analysis.

### 5.3 WEIGHT GENERATION

Learned weight generation is another promising application of our dataset. Recent works have explored generative models for neural network weights (Peebles et al., 2022; Schürholt et al., 2022a; Soro et al., 2024). By conditioning weight generation on phase and loss landscape metrics, as provided by our model zoos, future methods could produce models that are better aligned with specific target tasks or robustness criteria. Our dataset offers the diversity required to train such methods, moving beyond the relatively homogeneous datasets used in prior work.

## 6 DISCUSSION

**Limitations** In this work, the Phase Transition Model Zoos are limited to classification models in the computer vision domain. We focus on one domain to achieve better coverage for different computer vision datasets and architectures. Our work presents itself as a first step to make model zoos comprehensively cover phase transitions for a variety of applications, and we leave its extension to other tasks and domains for future work.

**Conclusion** The Phase Transition Model Zoos represent the largest structured collection of models annotated with detailed loss landscape metrics. With it, we provide the research community with a powerful tool to explore and benchmark neural network performance across different phases. By systematically covering phase transitions, it allows the study of robustness, generalization, and failure modes of deep learning methods in a much more nuanced, comprehensive and reliable way.

We demonstrate the relevance of phase transitions by identifying phases in experiments on fine-tuning, transfer learning, pruning, ensembling, and weight averaging. We show that these phases significantly affect performance and that their impact varies from one method to another, offering valuable insights beyond conventional performance metrics.

With this work, we encourage the ML community to leverage phase awareness in their evaluations, moving beyond single-point performance metrics and toward a deeper understanding of model behavior. Our dataset offers a foundation for advancing methods in population-based training, model analysis, and weight generation, contributing to more robust and generalizable machine learning models.

ACKNOWLEDGMENTS

Redacted for double-blind submission.

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

# A    DATASET DOCUMENTATION

A sample of the phase transition model zoo dataset can be downloaded anonymously from `https://drive.proton.me/urls/V2E66KY0JM#Pq5M06URN4EN`. Further instructions to explore, visualize or use the zoos can be found in the supplementary material and the corresponding `README.md`.

## A.1    MODEL ZOO CONTENTS

In the main paper, we described the generation of the model zoos as well as explored their performance and phase information. Here, we detail the contents of the datasets. A model zoo contains a set of trained Neural Network models. For each of the zoos, we fix architecture and task combinations and introduce variations in temperature-like and load-like parameters. We realize temperature variations by varying the batch-size, and load variations by varying the model width. We chose the training parameters and variation range such that the phases and phase transitions described by Yang et. al (Yang et al., 2021) can be observed. We repeat each temperature-load combination with seeds $\{1, 2, 3\}$ to compute loss landscape metrics and get robust results.

For every model sample, there are model state checkpoints at intervals throughout training. The checkpoints are in PyTorch format, which uses pickle to save ordered dicts. We will provide code to convert the checkpoints to framework-neutral file formats. We annotate these samples with performance metrics (training and test loss and accuracy), as well as the loss landscape metrics (hessian eigenvalues, Bézier mode connectivity, CKA similarity). We add additional results like model averaging performance, where applicable to individual models. The model zoos are generated with `ray.tune` [1] and largely follow their experiment structure. Each model in a population is contained in one folder. Checkpoints are kept in subfolders for the corresponding epochs. Each model is annotated with a `config.json` file to re-create the model exactly. Performance metrics are tracked for every epoch and saved in a `results.json` file for every model. For a subset of epochs, we add loss-landscape metrics. All model zoos contain full meta-data configs and self-contained Pytorch code, s.t. they can be re-instantiated exactly, re-trained, or fine-tuned. All code to train grids, evaluate, compute loss landscape metrics and model averaging is available alongside the data. Further, we provide code to i) recreate the model zoo datasets, ii) compute loss-landscape metrics, iii) load the models, iv) re-create the figures in the main paper. In order to allow easy use of the dataset, we plan to make adequate PyTorch dataset classes available upon publication.

This section will be updated upon dataset publication. Indeed, several statements are intentionally left vague as of now. Our dataset is large, and will require a careful choice on what to include in order to balance the dataset utility with its size. This will influence, in particular, the number of checkpoints we include per model.

## A.2    MODEL ZOO GENERATION

We generated the dataset for common computer vision tasks and architectures to maximize applicability to the community. We fixed the load-temperature grids by exploring the boundary cases first and establishing the presence of phase transitions, then filling in more resolution. We chose batch size as the temperature-like hyperparameter, and model width as the load-like hyperparameter: they are easy to vary and close to the practice. The amount of data is usually fixed, and learning rates are often scheduled and kept non-constant. The full list of model zoo hyperparameters is given in Table 1. We used Random Cropping, horizontal flipping and random rotations for all model zoos. Training ViTs on CIFAR100 required stronger data augmentation to achieve competitive performance. Therefore, we have applied a combination of random cropping, random erasing, color jitter, and RandAugment (Cubuk et al., 2019). After the initial tuning of the grids, the training of the model zoos was done on 16 DGX H100 GPUS in 20 days. The computation of loss landscape metrics was performed on the same hardware in 7 days.

---

[1] `https://docs.ray.io/en/latest/tune/index.html`

Table 1: Full list of hyperparameters of the model zoos. Variations between models are indicated by {...}. Width indicates the width of the first residual block. From that, we follow the same scaling factor as the standard ResNet.

| Base Architecture | ResNet-18, ResNet-50 | ViT |
|---|---|---|
| Datasets | SVHN, CIFAR10, CIFAR100, TinyImagenet | CIFAR10, CIFAR100 |
| Activation | ReLU | ReLU |
| Initialization | Kaiming Uniform | Kaiming Uniform |
| Optimizer | SGD | ADAMW |
| Learning Rate | 0.1 | $6e-3$ |
| Momentum | 0.9 | |
| WD | $5e-4$ | CIFAR10: $5e-4$. CIFAR100: $5e-2$ |
| LR Schedule | OneCycleLR with Cosine Annealing | OneCycleLR with Cosine Annealing |
| Width | 2, 4, 8, 16, 32, 64, 128, 256 | 6, 12, 24, 48, 60, 96, 192 |
| Batch Size | 8, 16, 32, 64, 128, 256, 512, 1024 | 8, 16, 32, 64, 128, 256, 512 |
| Seeds | 0, 1, 2 | 0, 1, 2 |

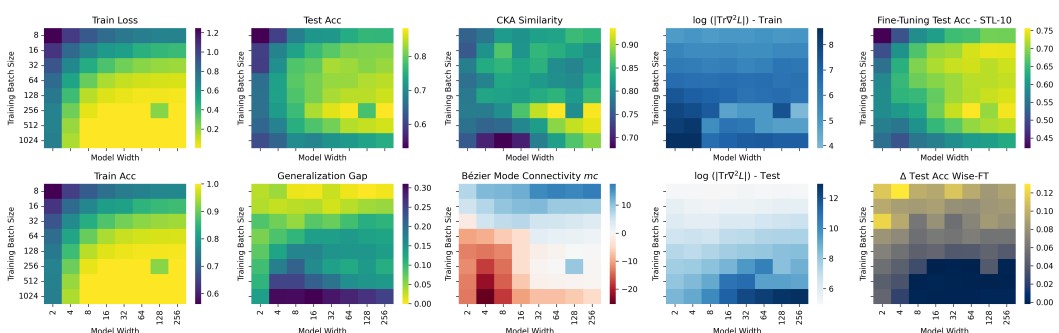

Figure 9: Weight averaging results for the CIFAR-10 ResNet-18 model zoo, showing distinct phase transitions in performance and loss-landscape metrics.

## A.3 MODEL ZOO EVALUATION

In this section, we test the general validity of the trained models as representatives of real-world models in a structured dataset. An overview of the models at the end of training is given in Table 2. The results confirm that models are trained to competitive performance for their respective size. More nuanced information on the distribution of model performance on the temperature-load grid is shown in Figures 9 through 18. Similar to previous work, the zoos show distinct low train-loss regions, with smaller embedded regions within that generalize well. Test performance generally improves with decreasing load (increasing width), with a distinct peak phase where temperature and load are low enough, but not too low. The generalization gap correspondingly shows a superposition of both patterns. Further applications or loss landscape metrics likewise show clear phase transitions.

Table 2: Conventional Performance Metric Distribution of Model Zoos.

| Model | Data | Train Loss $\mu \pm \sigma$ [min,max] | Test Loss $\mu \pm \sigma$ [min,max] | Train Acc $\mu \pm \sigma$ [min,max] | Test Acc $\mu \pm \sigma$ [min,max] | GGap $\mu \pm \sigma$ [min,max] |
|---|---|---|---|---|---|---|
| ResNet18 | SVHN | 0.10±0.11 [0.00,0.38] | 0.15±0.04 [0.11,0.27] | 0.97±0.03 [0.88,1.00] | 0.96±0.01 [0.92,0.97] | 0.01±0.02 [-0.04,0.03] |
| ResNet50 | SVHN | 0.06±0.07 [0.00,0.24] | 0.14±0.02 [0.11,0.18] | 0.98±0.02 [0.93,1.00] | 0.97±0.01 [0.95,0.97] | 0.01±0.02 [-0.02,0.03] |
| ResNet18 | CIFAR10 | 0.08±0.19 [0.00,0.66] | 0.67±0.34 [0.32,1.98] | 0.97±0.06 [0.77,1.00] | 0.82±0.08 [0.65,0.91] | 0.16±0.07 [0.04,0.35] |
| ResNet50 | CIFAR10 | 0.04±0.09 [0.00,0.52] | 0.60±0.30 [0.27,1.69] | 0.99±0.03 [0.82,1.00] | 0.84±0.07 [0.64,0.92] | 0.15±0.06 [0.05,0.33] |
| ResNet18 | CIFAR100 | 0.45±0.79 [0.00,2.48] | 2.02±0.54 [1.24,3.89] | 0.88±0.21 [0.35,1.00] | 0.53±0.12 [0.29,0.69] | 0.35±0.15 [0.01,0.67] |
| ResNet50 | CIFAR100 | 0.35±0.68 [0.00,4.61] | 1.78±0.55 [1.18,4.61] | 0.91±0.18 [0.01,1.00] | 0.57±0.11 [0.01,0.70] | 0.34±0.14 [-0.01,0.67] |
| ResNet18 | TI | 1.20±1.06 [0.01,3.42] | 1.91±0.48 [1.29,3.22] | 0.71±0.25 [0.23,1.00] | 0.55±0.12 [0.26,0.70] | 0.16±0.15 [0.03,0.41] |
| ResNet50 | TI | 1.05±0.96 [0.00,3.55] | 1.85±0.51 [1.21,3.63] | 0.74±0.22 [0.21,1.00] | 0.57±0.11 [0.22,0.72] | 0.17±0.15 [-0.02,0.49] |
| VIT | CIFAR10 | 0.77±0.83 [0.00,2.18] | 1.72±0.45 [0.71,2.96] | 0.71±0.31 [0.17,1.00] | 0.59±0.23 [0.10,0.82] | 0.13±0.10 [-0.01,0.27] |
| VIT | CIFAR100 | 2.96±0.94 [1.21,4.32] | 2.72±0.82 [1.74,4.15] | 0.37±0.24 [0.06,0.88] | 0.43±0.22 [0.09,0.72] | -0.05±0.07 [-0.13,0.16] |

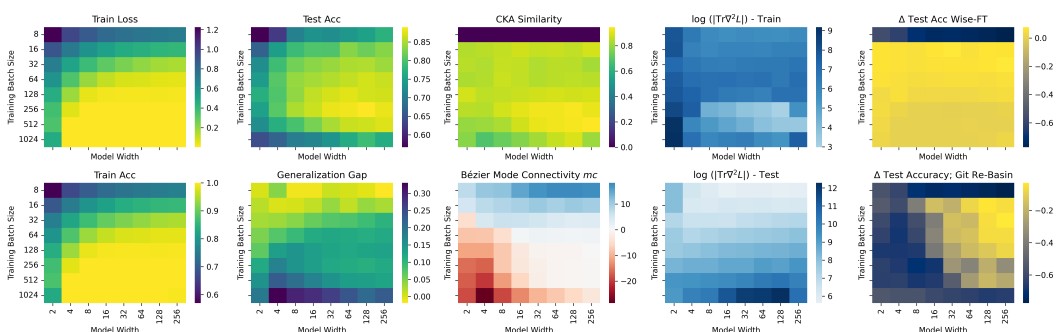

Figure 10: Weight averaging results for the CIFAR-10 ResNet-50 model zoo, showing distinct phase transitions in performance based on averaging strategy.

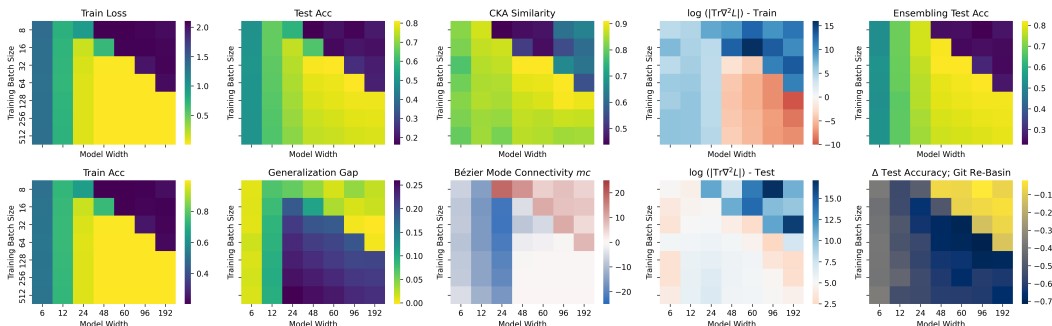

Figure 11: Weight averaging results for the CIFAR-10 ViT model zoo, showing distinct phase transitions in performance based on averaging strategy.

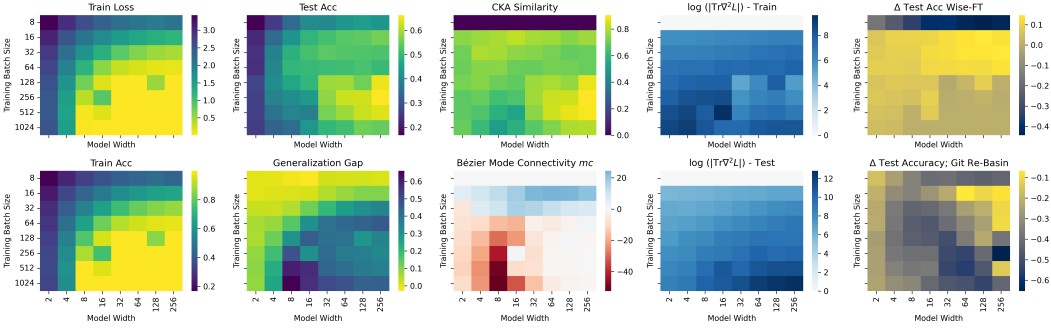

Figure 12: Weight averaging results for the CIFAR-100 ResNet-18 model zoo, showing distinct phase transitions in performance and loss-landscape metrics.

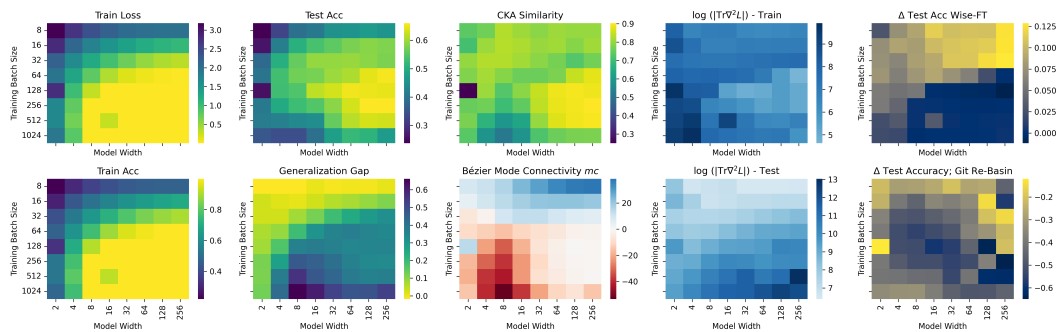

Figure 13: Weight averaging results for the CIFAR-100 ResNet-50 model zoo, showing distinct phase transitions in performance based on averaging strategy.

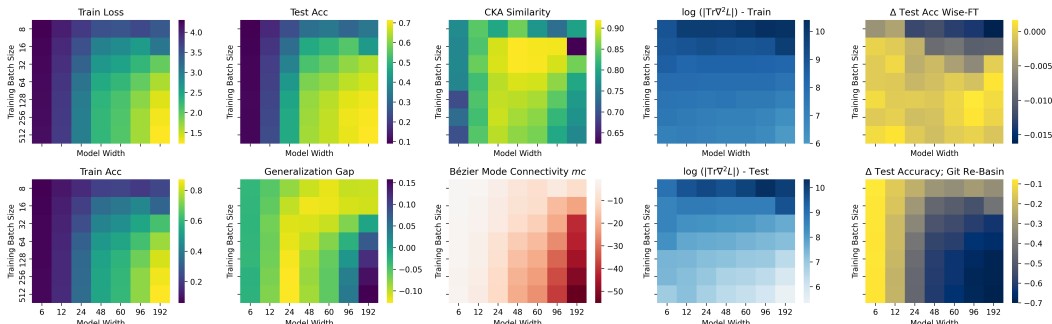

Figure 14: Weight averaging results for the CIFAR-100 ViT model zoo, showing distinct transitions in performance based on averaging strategy.

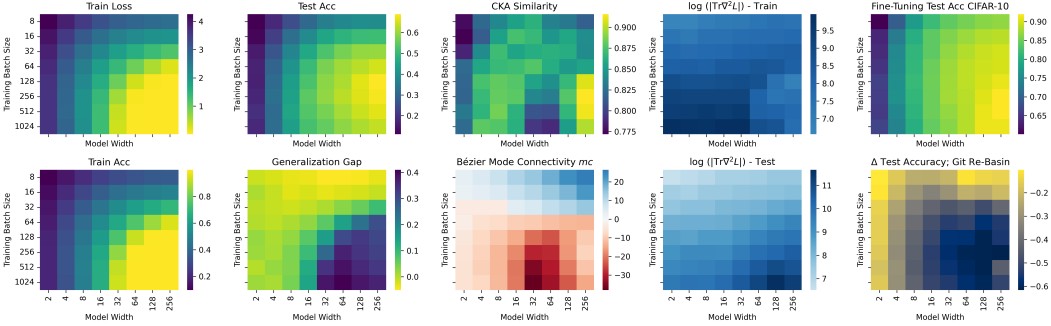

Figure 15: Weight averaging results for the Tiny-Imagenet ResNet-18 model zoo, showing distinct phase transitions in performance and loss-landscape metrics.

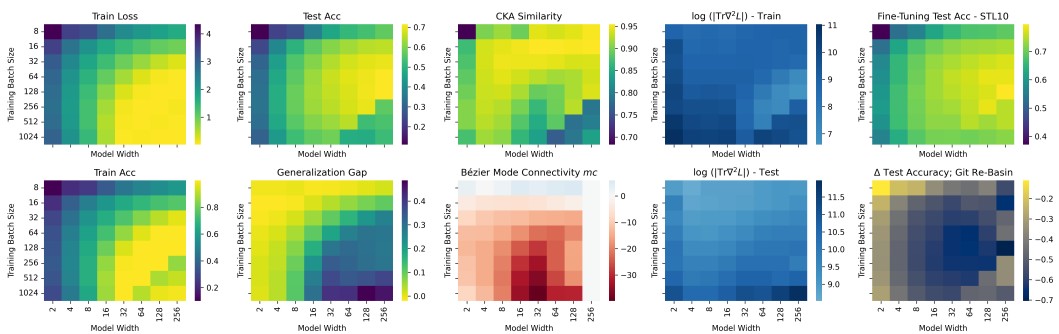

Figure 16: Weight averaging results for the Tiny-Imagenet ResNet-50 model zoo, showing distinct phase transitions in performance based on averaging strategy.

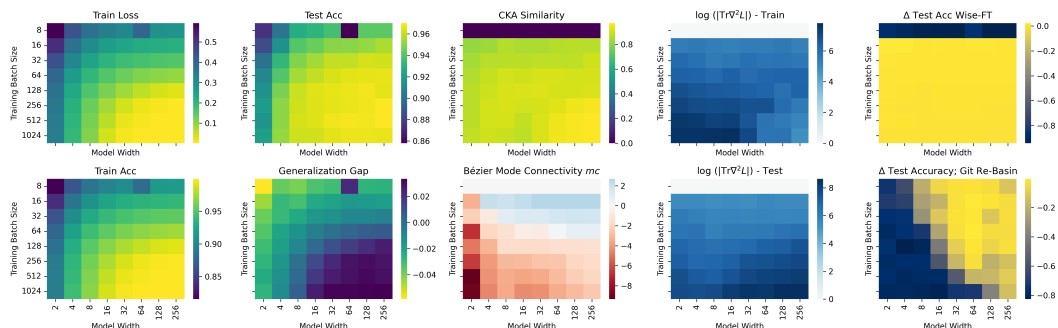

Figure 17: Weight averaging results for the SVHN ResNet-18 model zoo, showing distinct phase transitions in performance and loss-landscape metrics.

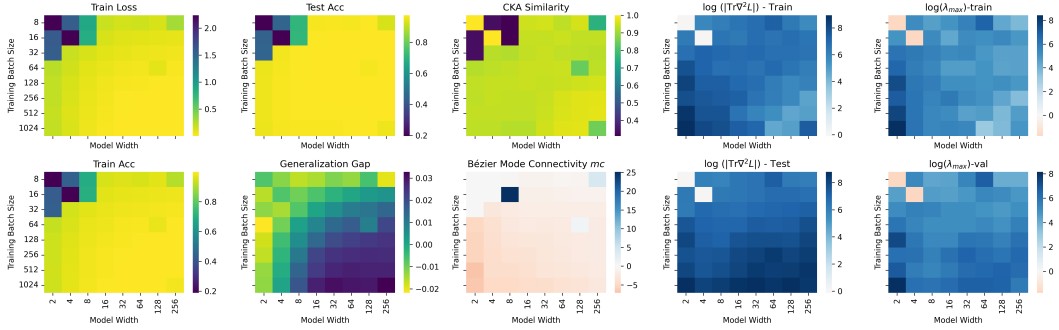

Figure 18: Weight averaging results for the SVHN ResNet-50 model zoo, showing distinct phase transitions in performance based on averaging strategy. Note that extreme values for low width and batch size distort the appearence of the phase plots.

## A.4   INTENDED USES

The dataset is a repository of trained deep-learning models with phase transitions. It is mainly intended to study phase transitions on populations of neural network models. For every model, we include multiple checkpoints, representing different training epochs, to allow for the study of the training procedure. We also provide loss landscape metrics, to allow researchers to relate their findings with the structure of the loss landscape. The dataset is intended to allow researchers to **(i)** identify phases in different model properties or applications like the weight averaging examples in the main paper; and **(ii)** evaluate existing methods that rely on pre-trained models systematically on models of different phases, to get a better understanding under which conditions methods can be expected to perform well. Further examples of applications of our dataset are presented in our publication: model training, model property prediction, model generation, model combination, etc.

Please note that this dataset is intended for research on populations of models, not to further improve performance on specific computer vision tasks directly. The models in our zoo were selected for their diversity in phases, not optimized for performance on their specific datasets; there may exist generating factors combinations achieving better performance with similar architectures.

The dataset is entirely synthetic and does not contain personally identifiable information or offensive content. Authors bear all responsibility in case of violation of rights.

## A.5   HOSTING, LICENSING, AND MAINTENANCE PLAN

The dataset will be made publicly available and licensed under the Creative Commons Attribution 4.0 International license (CC-BY 4.0). **We refrain from publishing the dataset during the review process to ensure double-blind reviewing. The dataset will be published after the completion of the review process, for the camera-ready version. We provide representative samples of the dataset to the reviewers via anonymous uploads. We will incorporate reviewer feedback for the published version of the dataset.** This dataset documentation will be updated to include the corresponding links and information.

We plan on using Zenodo for data hosting. It provides persistent identifiers (DOI), long-term hosting guarantees for at least 20 years, and dataset versioning. We will collect information on the dataset, links to the data, code, and projects with the dataset on a hosted website. Upon publication, code to use the datasets will be made available, including multi-purpose dataset metadata like croissant. We will maintain the datasets if necessary. We further plan on extending the dataset towards more architectures, tasks, and domains, and invite the community to engage.

Code to recreate, correct, adapt, or extend the datasets will be provided in a GitHub repository, such that maintenance can be taken over by the community at need. The GitHub repository allows the community to discuss, interact, add, change, or fork the code.

