# OpenReview forum: "Model Zoos for Benchmarking Phase Transitions in Neural Networks"
_ICLR.cc/2025/Conference — Submitted to ICLR 2025_

### Official Review · Reviewer_Qvjj · 2024-10-31

**Soundness:** 1
**Presentation:** 2
**Contribution:** 2
**Rating:** 3
**Confidence:** 3

**Summary:**

This work presents model zoos as a contribution to the study of phase transitions of neural networks (NNs) occurring in different scenarios. The authors recognize the necessities of phase transitions in NNs for a deeper understanding of deep learning, which could bridge the gap of our understanding of NNs. Based on the loss landscape taxonomy (loss landscape metrics and the categorized phases) of previous works, the authors annotated performance and loss landscape metrics across a large population of pre-trained models to identify such phases in their corresponding loss landscapes. The study empirically shows that phases robustly emerge within each model zoo population, and demonstrates that it also relates to the phases of different NN training methods or applications.

**Strengths:**

Presenting a large scale of model zoos for phase transition of NNs is indeed strength of the work. I agree with the authors that the model zoo about this scale could be valuable to researches of understanding the fundamental behavior of NNs.
While related works have already shown the existence of different phases based on loss landscape metrics, this work strenghten the previous works by building a diverse population of model zoos.

Section 3 shows distinct phases emerge across model population, and Section 4 relates these phases of the pre-trained models with diverse NN training methods. These heavy empirical results are the strengths of the paper.

**Weaknesses:**

While I highly appreciate the authors work for building a large scale of model zoos to the community those who are interested in the foundational understanding of NNs, I think that the following points should not be left undiscussed.
1. First of all, while the authors claim 'phase transitions' or 'phases' of NNs, I think this term should be used with more strictness. I understand the phase transition of NNs couldn't be nicely written with mathematical (or physical) rigor. The previous study which seems to provide the basic setting of this work [1] is published almost 5 years ago, and I would expect to give more theoretical aspects of this empirical findings. For instance, what is the order parameter in this scenarios? Is it a discontinuous or continuous transition? Of course it is indeed a challenging problem, but providing a theoretical discussion and its corresponding empirical evidence would highly improve the concreteness of this work. Also, while the answers of my sample questions could be interpreted with the provided experimental results, but I expect adding a discussion about this would add more clarification to the general audience.
2. Closely related to the first point, where is the phase 'transition'? While it is helpful to discriminate distinct phases across the loss landscape, in the notion of phase transition, investigating the transition between two phases should be accompanied in this empirical study. For example, between Phase IV-A and Phase IV-B, if temperature corresponds to the control parameter and CKA corresponds to the order parameter (my tentative guess), I would expect a empirical results showing the phase transition curve along these two parameters.
3. In the Figure 4 and 5, I'm curious about the model similarity (model distance or CKA) between the pre-trained model and after fine tuning or transfer learning. I couldn't find the exact settings for the fine tuning or the transfer learning both in main text and supplementary materials, so if I might be so bold as to predict, I don't expect the model before and after transfer learning should differ that much, and based on that, it does not seems surprising. Providing the model similarity before and after fine tuning transfer learning, and the different phase plots across different transfer learning settings (e.g. training only the last block, last two blocks, and so on.) could be helpful.
4. Similar with the third point, ablation experiment with respect to pruning ratio could help support the authors claims.


[1] Yaoqing Yang, Liam Hodgkinson, Ryan Theisen, Joe Zou, Joseph E Gonzalez, Kannan Ramchandran, and Michael W Mahoney. Taxonomizing local versus global structure in neural network loss landscapes. In Advances in Neural Information Processing Systems, volume 34, pp. 18722–18733. Curran Associates, Inc., 2021.

**Questions:**

My main concerns are stated in the weaknesses part above. For the questions, I've found several typos and possible miswritten texts that made me confused to follow the paper. I would like to ask some of ambiguities regarding those, please correct me if I'm wrong.
1. Near line 305, how does Figure 9 and 11 relates the phase of the pre-trained model? In the supplementary materials, the captions for Figure 9 and 11 says weight averaging.
2. Same for line 332; Figure 15 and 16, line 385; Figure 11.
3. Due to the minor point 1 and 2, maybe the captions of Figure 9-17 (line 421) which are intended to show more examples of weight averaging could be wrong I believe?
4. In Figure 6, the caption describes that the figure shows (top) before pruning and (bottom) after pruning. However the top subfigure shows the plot for mode connectivity. Which description is correct?
5. It would help the general audience to understand the setting of the experiments regarding weight averaging if Wise-FT method is mentioned in the main text (which happens to be not, while Git Re-basin have been mentioned).

---

> ### Author Response · Authors · 2024-11-22
> **Response to Reviewer Qvjj**
>
> We would like to thank reviewer Qvjj for their feedback. We appreciate that they find our dataset `large scale` which as a consequence `could be valuable to researchers understanding the fundamental behavior of NNs`. This is ultimately what we wanted to provide as a dataset.
>
> However, the review raises several critiques. Some of these appear to us to be based on the misconception of our paper as a methods paper. As we elaborate in our response to all reviewers, it is a dataset submission that does not aim at making method contributions nor establishing new findings. In the following, we respond to the individual points.
>
>
> ## Rigor in Phase Transitions
> We wholeheartedly agree with the reviewer. We find the concept of phase transitions both relevant and exciting, to build a deeper understanding of neural networks but also, e.g., explain challenges in reproducibility of methods. While there has been some work on the domain, which we summarize in Section 2 of the submission, more work should be done, specifically to address the questions the reviewer raised. We understand our dataset as the necessary basis for such research and hope our dataset can enable that research, s.t. the questions the reviewer and we have can be answered.
>
> ## Discussion of Order and Control Variables
> We thank the reviewer for the feedback. Previous work on statistical physics has suggested that `load-like` and `temperature-like` hyperparameters are the control variables of this system, which we discuss in Section 2 of the submission. Following this notion, load-like parameters can be instantiated differently, but changing different load-like parameters will have the same effect of changing the load. The same goes for the temperature-like parameters. We borrow this notion, as it is useful to simplify the landscape of hyperparameters to the qualitative axis and we have no indication that it may be fundamentally wrong.
>
> The phase transitions are observed in order variables. Here, we want to highlight that all metrics we observe, be it test accuracy, training loss, hessian, connectivity, or downstream applications like pruning or transfer learning performance can be understood as order variables. Also, our dataset is not intended to cover loss-landscape metrics. It is intended to cover phase transitions, which happen to coincide with some loss-landscape metric phase transitions. We will revise the submission to include that discussion.
>
> ## Grid Resolution
> The reviewer notes that based on our experiments establishing the sharpness or smoothness may be challenging. We agree with that. Research to identify the location and sharpness of specific transitions may have to interpolate between the nodes in our grid - but they benefit from the grid as a starting point. We strongly believe there is value in that knowledge that facilitates further research on that domain.
>
> ## Ablation studies over pruning
> We appreciate the question. We have performed ablation studies w.r.t. pruning ratios, which yield different distributions of the phases but generally similar trends. However, we want to highlight that the goal of our paper is not to establish any particular relation between control to order parameters, which is out of scope for a dataset. In fact, the particular aspect of pruning has been the topic of  [1]. Instead, our intention for these experiments is to illustrate the fact that there are phase transitions in many different aspects of neural network methods. As we argue in our response to all reviewers, it is for that reason that phase transitions should be considered. Our dataset can be used to evaluate for phase transitions w.r.t pre-trained models to that end. We will revise the draft to make that clearer.
>
> ## Questions
> We thank the reviewers for the pointer to the caption of the figures in the appendix. In the heat of the submission we missed the captions, they in fact show different metrics, not just weight-averaging. We will fix that in the draft.
>
> ### Q1 + Q2.
> Figure 9 at the top right shows fine-tuning results from CIFAR-10 ResNet-18 zoo to STL-10. Our apologies for referencing Figure 11 to fine-tuning. We had to make a choice of what results to show for each zoo, and that figure changed after the reference was included. We will fix that reference.
>
> ### Q3.
> Here, too, our apologies for the error in the captions. We changed the figure last minute but apparently missed updating the caption.
>
> ### Q4.
> We thank the reviewer for the pointer. Wise-FT is in fact weight-averaging over epochs. We will add that information to the draft.
>
>
> [1] A Three-regime Model of Network Pruning, Zhou et al., ICML 2023.
>
> ---
>
> Once again, we would like to thank the reviewer for the feedback and hope our response could address the questions. We look forward to the discussion.

---

> > ### Comment · Reviewer_Qvjj · 2024-11-25
> >
> > Thank you for clarifying the objective and goals of this paper (in the area of Dataset and Benchmark) in your response. While I partly agree with the authors' claims regarding the scope of the paper—specifically, (1) addressing the ablation experiments I proposed (weaknesses 3 and 4), and (2) considering the grid resolution (which, although not explicitly mentioned in my feedback, I acknowledge)—I respectfully disagree with the notion that weaknesses 1 and 2 should be left undiscussed within the scope of this paper.
> >
> > The authors note that this work is a dataset submission and does not aim to contribute new methods or establish new findings. While I do not have a strong objection to this, I believe that if the paper is titled "... Benchmarking Phase Transitions in Neural Networks" and the main message (beginning with the abstract) suggests a focus on 'phase transitions in neural networks,' it should at least include primary results that support to open up such a discussion (i.e., addressing weaknesses 1 and 2). This would help justify the claim that the paper provides a benchmark for this topic.
> >
> > Once again, thank you to the authors for their prompt response. I will maintain my score accordingly.

---

### Official Review · Reviewer_JhyP · 2024-11-03

**Soundness:** 2
**Presentation:** 2
**Contribution:** 2
**Rating:** 3
**Confidence:** 3

**Summary:**

This paper introduces a dataset of trained models with multiple checkpoints across 10 combinations of ResNet and ViT architectures, trained on CIFAR-10, CIFAR-100, SVHN, and Tiny ImageNet. Each model is trained over a grid of varying batch and model sizes (interpreted as the temperature-load landscape). Through analysis of loss landscape metrics, the paper identifies distinct phases and phase transitions within this landscape. The main result demonstrates that phases and phase transitions exist also in a range of applications such as pruning, fine-tuning, ensembling, transfer learning, and weight averaging, suggesting that the presence of distinct phases is a prevalent phenomenon.

**Strengths:**

The paper demonstrates the existence of phase transitions in a more diverse set of models/datasets compared to prior work as well as on a variety of neural network methods including pruning, fine-tuning, ensembling, transfer learning, and weight averaging.

The work encourages evaluating methods across different model phases and phase transitions, suggesting that examining model behavior systematically across various regimes can lead to more robust evaluation methods than single-point evaluations.

In particular cases, the analysis of the phases provides insight on how certain components of the training pipelines such as learning rate decay and data augmentation impact the phases.

**Weaknesses:**

The paper primarily extends earlier work on phase transitions in neural networks to a broader range of datasets, models and methods but lacks substantial novelty.

Despite Section 5 outlining possible uses for the Model Zoo dataset, no concrete application is provided to demonstrate its practical relevance. Including a simple concrete example would be highly beneficial in demonstrating the utility of the dataset.

The scope of the Model Zoo dataset is limited by the use of relatively small datasets and models, excluding e.g larger and more widely used datasets such as ImageNet.

**Questions:**

- Could you please elaborate on how Model Zoos specifically "help researchers systematically identify phase transitions in their methods", as suggested in the paper?

- In general, how can researchers utilize the provided Model Zoo to benefit different methods beyond those directly included in this work?

- Could you please elaborate on the statement in lines 258-259: "... decaying the learning rate by 1e4 under cosine annealing increases the area of Phase IV (well-trained regime) while reducing the presence of Phase II …". It is not quite clear to me how this conclusion is made?

- Why does line 242 say models are trained using random seeds {1, 2, 3}, but in Table.1 seeds are reported to be {0,1,2}?

- In Figure.5 the dataset used in the bottom grid, CIFAR10, doesn't match the dataset specified in the caption STL-10.

---

> ### Author Response · Authors · 2024-11-22
> **Author Response to Reviewer JhyP**
>
> We thank reviewer JhyP for the feedback. We are thankful that the reviewer found our illustration of phase transitions as a general concept convincing.
>
> Before we respond to the individual points, we want to clarify that our submission is a dataset submission and does not aim for methodological novelty. As is our impression of the reviewer, we find the research on phase transitions both interesting and relevant. That research generally requires carefully generated model datasets like ours. As we are not aware of any large-scale dataset to that effect, we present our dataset to facilitate further work in that domain. If the reviewer is aware of other datasets on phase transitions of the same size or scope, we would gladly discuss the relation and compare our work with it. As is, we believe that our dataset could facilitate research and would benefit the community and therefore merits publication.
>
> ## Scope of the dataset (also Questions 1 and 2)
> We discuss the scope of our dataset in detail in our response to all reviewers. We agree with the reviewer that including more/larger models/datasets with more variations is always desirable, but our resources are not unlimited. Between breadth or depth, we found it to be more interesting to establish that phase transitions generalize between more different datasets, architectures, tasks, and domains, rather than investing the resources in a single grid which might show the particularities of this one setup.
>
> ## Applications of the dataset (also Questions 1 and 2)
> This point, too, is discussed in our response to all reviewers. Datasets like ours are a necessity for research on phase transitions, as well as for population-based methods. This research may consider identifying or describing phase transitions in various metrics or applications. It may also use the knowledge of existing phase transitions to diagnose models or towards hyperparameter optimization.
>
> For example, previous work has demonstrated that loss landscape metrics predict phase information [1]. Their approach provides a systematic framework for identifying phases in model performance through the analysis of loss landscape metrics. Specifically, they identify phases by combining key loss landscape properties, where the optimal phase is characterized by low sharpness, high CKA similarity, and zero mode connectivity.
> That means, for any given computer vision model, whether it is a ResNet with suboptimal load or temperature or a pruned model, one can evaluate its phase by measuring these loss landscape metrics. Comparing these metrics to the optimal phase allows us to determine whether the model is in a desirable state or if adjustments are required.
>
> As shown in [2], the knowledge of phase layout and its relation to loss landscape metrics can also be used to diagnose failure modes toward hyper-parameter optimization. Rather than just using training and validation losses, the loss landscape metrics can be used to localize a current model on the phase plot and navigate quickly toward optimal phases. Our dataset is a resource to both understand the layout of phases, but also to develop and evaluate phase-navigation methods across multiple architectures and datasets.
>
>
> ## Question 3:
> Previous work on phase transitions largely used constant learning rates to train grids of models. Due to their omnipresence in research and practice, we decided to use a learning rate scheduler. Comparing grids without an LR scheduler, as well as the phase distribution before the learning rate is fully decayed shows distinct and relatively balanced phases II and IV. Decaying the learning rate results in a very large phase IV, which may explain why LR schedulers are so successful - it seems to make training less sensitive to mis-parameterized load/temperature. Incidentally, this is the type of insight that we call for by evaluating phase transitions.
>
>
> ## Question 4:
> Apologies for the error in the caption. We will fix them in the revision.
>
> [1] Yang et al., Taxonomizing local versus global structure in neural network loss landscapes, NeurIPS 2021.
> [2] Zhou et al., MD tree: a model-diagnostic tree grown on loss landscape, ICML 2024.
>
> ---
>
> We hope we could address the reviewer’s questions. We look forward to the discussion.

---

> > ### Comment · Reviewer_JhyP · 2024-11-25
> >
> > I thank the Authors for their response. I appreciate the insights offered by the Model Zoo dataset and the paper’s emphasis on evaluating methods across phase transitions rather than single-point estimates. I also understand that proposing novel methods is not the focus of this work.
> >
> > However, it is unclear how this particular dataset can be directly applied to alternative methods. For instance, the Authors suggest that population-based methods for hyperparameter tuning could benefit from incorporating loss landscape metrics to guide the search more effectively than relying solely on validation accuracy. While this statement might be valid, it is not clear to me how the Model Zoo dataset specifically could be used in practice in such an application.

---

> > > ### Author Response · Authors · 2024-11-25
> > >
> > > We thank the reviewer for their response and are glad our previous response provided insights.
> > >
> > > To the specific question: research towards hyper-parameter tuning is a good example. [1] demonstrated that using loss-landscape metrics can help diagnose failure modes of models, i.e. whether a model is too small or trained on too little data. They also demonstrate that loss landscape metrics outperform conventional validation metrics to that end. While not done in [1], using loss-landscape metrics to tune load-like and temperature-like parameters on a new task or with a new architecture is a relatively straight-forward extension.
> > >
> > > Developing and evaluating such hyper-parameter optimization methods requires trained models with loss-landscape or phase annotation across multiple datasets and architectures, like ours. Researchers working on this domain therefore benefit from available datasets, as it saves them from generating their own. Sharing a common dataset also helps evaluating different methods against each other.
> > >
> > > We hope that example illustrates applications that would benefit from our dataset. Once again, thank you for the feedback and welcome any further questions.
> > >
> > > [1] Zhou et al., MD tree: a model-diagnostic tree grown on loss landscape, ICML 2024.

---

> > > > ### Comment · Reviewer_JhyP · 2024-11-25
> > > >
> > > > I thank the Authors for their prompt response regarding application of the Model Zoo dataset in population-based hyperparameter optimization. However, on a new task, I expect the benefits to be very limited as this depends on how similar the models under study are to the models considered in Model Zoos. Based on this, I am inclined to maintain my rating.

---

> > > > > ### Author Response · Authors · 2024-11-25
> > > > >
> > > > > We likewise thank the reviewer for the prompt response and appreciate the discussion.
> > > > >
> > > > > We agree with the reviewer that exploring or even covering the hyper-parameter space appears a daunting task. This is the precise reason why we feel the concept of phases is powerful: it describes qualitatively similar models, regardless of how they were trained to reach that particular phase. Previous work as well as our own dataset demonstrate that the phases generalize different architectures, datasets, model sizes and choices for load-like and temperature-like parameters. Since our datasets cover the known phases, they represent a broad range of model behavior of computer vision models. We therefore believe they are well-suited to develop and evaluate methods.

---

### Official Review · Reviewer_TH1T · 2024-11-05

**Soundness:** 2
**Presentation:** 3
**Contribution:** 2
**Rating:** 5
**Confidence:** 3

**Summary:**

The paper presents a structured dataset of trained neural networks to facilitate the study of phase transitions in neural networks. Specifically, the authors trained a collection of two types of ResNet models (18 and 50) on SVHN, CIFAR-10, CIFAR-100, and TinyImageNet as well as a ViT on CIFAR-10 and CIFAR-100. For each model and dataset, they trained a grid of models using 8 (7 for the ViT) different model widths (as an example of a "load-like" hyperparameter, determining the model capacity) and 8 (7 for the ViT) different batch sizes (as an example of a "temperature-like" hyperparameter, affecting the noise-level of training). Each training is repeated with 3 different random seeds for a total of >1800 trained models.
The dataset contains multiple checkpointed model weights at different training epochs, along with performance metrics (e.g. train/test loss & accuracy), and crucially, "loss landscape metrics". These metrics, such as the trace and largest eigenvalue of the Hessian, or mode connectivity measurements, help describe in which "phase" a neural network is (e.g. underfitted, undersized, etc.) and identify phase boundaries.

The authors then use their dataset to demonstrate the usefulness of phase information in four ML methods (fine-tuning, transfer learning, pruning, and ensembling). For example, they identify that "the best pre-trained model does not result in the best transfer-learned model" and that incorporating phase information can be beneficial.

**Strengths:**

- The paper curates a structured dataset of trained neural networks aimed specifically at phase transitions. The provided neural networks include meaningful observables, like Hessian statistics. Sharing this dataset could help facilitate research, by allowing researchers to directly use the dataset without spending the compute training a large number of models. In my opinion, the strength of the dataset is the addition of loss landscape metrics describing the network beyond the performance metrics.
- The paper is well-written and organized. It is structured clearly, describing the included metrics, and the process of generating the dataset, as well as a section describing potential applications.
- Phase transitions are an interesting lens to study neural networks. Having a comprehensive dataset is a great building block to accelerate research in this area. In particular, studying fine-tuning, transfer learning, pruning, and ensembling using phases is interesting and - at least to me - novel.

**Weaknesses:**

### Dataset has a limited scope

- As mentioned by the authors, the model zoo is limited to computer vision models and specifically image classification. Furthermore, it only uses (relatively) small-scale models and datasets. I appreciate that the paper acknowledges the domain limitation and that this current version of the dataset already required significant computational resources. However, I also believe that the usefulness and therefore impact of the work would increase significantly by considering other domains or model scales.
- By varying batch size and model width, the current dataset only explores a single temperature-like and load-like parameters. As a reader, I often asked myself whether a specific conclusion applies to all temperature or load-like parameters or is a specific property of batch size/model width. Having only one hyperparameter per type makes it impossible to disentangle the question and thus limits the usefulness of the dataset.
- Additionally, the "grid resolution" (i.e. number of batch sizes or model widths) is relatively low. I am aware that increasing the grid also substantially increases the computational costs. However, I am not sure whether the current resolution allows a sufficient distinction between smooth and abrupt changes, i.e. allow meaningful identification of phase transitions. In other words, phase transitions are basically about detecting sharp edges in the plots but a low-resolution or blurred plot makes it hard to detect edges. In comparison, Yang et al. (2001), seemed to use a significantly higher resolution.

### Phase transitions insights

I am unsure how robust the provided insights are and how well they generalize.
For example, the conclusion that "fine-tuned models exhibit clear phase transitions in their performance, and these phases overlap substantially with the phases of the pre-trained models" is only drawn from 2 or 3 examples (the section mentions Figures 4, 9, and 11, but Figure 11 doesn't have a "fine-tuning" subplot).
Similarly, the conclusion that "Choosing the best pre-trained model does not result in the best transfer-learned model" is taken from 3 examples (Figures 5, 15, and 16). However, in Figure 16, as far as I can tell, the best test accuracy is achieved at the same point as the best fine-tuning test accuracy (batch size 128 and 256 model width).
Additionally, the distinction between fine-tuning and transfer learning is unclear (see "Questions" Section) with all subplots being labeled "Fine-tuning Test Acc".

Similarly, the conclusion on pruning, "Models pre-trained in higher-temperature phases [...] tend to perform better post-pruning [...]" (line 358), seems to be based on a single example (Figure 6) (and even here, the batch size of 256 seems to be a bit of a counter-example).
For weight averaging, the authors state that "Averaging over epochs improves performance in early phases (I and II), where the Hessian trace is large" (line 422). However, Figure 14 seems to be a counter-example: Here, the $\Delta$ Test Acc Wise-FT is negative where the Hessian trace is large (i.e. top right corner).

Overall, I think the conclusions are not convincingly demonstrated. Perhaps showcasing the predicted power of the data could strengthen this. By only looking at the phase information, can you predict which model will have the best pruning/fine-tuning performance?

### Unclear applications for future research

This might be subjective and due to my research background but I have trouble grasping concrete applications of the dataset beyond further studying the dataset itself. For example, in line 105, the authors write, that the dataset "can help researchers systematically identify phase transitions in their methods, and comprehensively benchmark those". What type of methods does this refer to? How can the model zoo be used to identify phase transitions for new methods?
Despite the dedicated section (Section 5), I find it hard to see direct use cases of the dataset which wouldn't require extensive re-training.

**Questions:**

### Open Questions

- I am unfamiliar with CKA. Does a larger CKA indicate more similar representations (or less similar)?
- Do you perform any hyperparameter tuning, e.g. for the learning rate? You mention that "By varying the width, we achieve variations in model capacity without changing the architecture or having to adapt the training scheme" (line 238), but wouldn't changing the width require tuning the learning rate if not using something like muP?
- In line 256, do you mean "Phase IV-A" instead of "IV-B"?
- What is your distinction between fine-tuning (Section 4.1) and transfer learning (Section 4.2)? Depending on the community, the difference might be adapting to a new dataset vs. a new task. But here, in both cases, it is image classification no? And Figure 5, which belongs to the transfer learning section, also calls it "Fine-Tuning Test Acc".

### Plots

To me, some of the plots were confusing or missing information. Some (subjective)examples:

- Some plots are small and thus hard to read, e.g. the subplots in Figure 3.
- I think the plots would be improved if all subplots contained the phase annotation. For example, in Figure 3, only the subplot a has annotated phases and I would love to see the same separation in the other subplots. Same with the plots in the appendix, where no phase distinction is provided.
- Figure 1 is missing a color bar. Although primarily a qualitative plot, I believe a color would still be useful as the main argument is about the existence of phase transitions, i.e. abrupt changes in model behavior. Without a color bar, it is difficult to distinguish between an abrupt and a smooth change given the low grid resolution.
- Figure 6 seems to have the wrong subplot (or wrong caption) for the top subplot.
- What are subplots with the title "$\Delta$ Test Acc Wise-FT"? Judging from the context, I am assuming this is the weight-averaging strategy (i) as described in Section 4.5.

### Nitpicks

- Line 74: Is there a typo in "over different pre-trained models - model populations"?
- In line 90, you mention that the "model zoos contain a total of 1829 models". My calculation results in 1830 models, as follows: $(\underbrace{2}_{\text{ResNet models}} \cdot \underbrace{4}_{\text{ResNet datasets}} \cdot \underbrace{8 \cdot 8}_{\text{ResNet grid, e.g.~width and batch size}} + \underbrace{1}_{\text{ViT models}} \cdot \underbrace{2}_{\text{ViT datasets}} \cdot \underbrace{7 \cdot 7}_{\text{ViT grid, e.g.~width and batch size}}) \cdot \underbrace{3}_{\text{random seeds}}$. Did I misunderstand something?
- The paper uses inconsistent capitalization, i.e. "Testing Methods on Populations of Neural Networks" (line 70) vs. "Loss landscape metrics" (line 133).
- Line 286, there should be a space before the citation.

---

> ### Author Response · Authors · 2024-11-22
> **Author Response to Reviewer TH1T**
>
> We would like to thank Reviewer TH1T for the thoughtful and extensive feedback. We appreciate the level of detail and the time invested in the review. All the more, we are grateful the reviewer found our submission `well-writtten and organized`, the dataset `comprehensive` and a `great building block for future research`, and the topic of phase transitions an `interesting lens to study neural networks`. As a dataset, this is what we aimed it to be, and are glad that, despite the low score, the reviewer agrees with us.
>
> Nonetheless, in the following, we address the reviewer’s points and hope to supply sufficient reason to reconsider the score.
>
> ## Dataset Scope
> We discuss the dataset scope in our response to all reviewers but elaborate here further.
>
> Regarding the instantiations for load and temperature, we agree with the reviewer that further investigations would be interesting, but ultimately we found the evaluations of different datasets, architectures, and domains more interesting. Previous work [1] has evaluated different realizations, comp. e.g. appendix D5 and D6. They conclude that the results are very similar. Further, we understand the phases as distinct types of models. In this dataset, we are more interested in achieving balanced coverage of known phases than in different ways of achieving them. That said, if our dataset sparks future research on the equivalences of different temperature or load parameters, we count that as success regardless of the outcome and welcome any addition to the dataset.
>
> Naturally, a higher resolution may likewise be desirable. However, we believe the dataset is valuable as is for two reasons. First, it allows us to uncover broad trends, which is preferable to not observing any trends. Further, it helps locate the different phase transitions of different applications and allows researchers to increase the local resolution of the grid where necessary. We believe this is a big advantage over starting from scratch. Again, we do not see our dataset as the do-all end-all dataset, but rather a first step that allows meaningful research and will hopefully be extended by us and the community.
>
> ## Dataset Applications
> We discuss the applications for our phase transitions dataset in the response to all reviewers. We hope that illustrates use-cases for our dataset, and are happy to elaborate further or give more examples. We will revise the section in the submission to be more explicit.
>
> ## Phase Transition Insights
> First, we would like to thank the reviewer for the feedback. It appears that in the rush for submission, we made mistakes, particularly in the captions. We will fix these in the revision.
>
> With the examples of phase transitions, we wish to illustrate the point that phase transitions are a general phenomenon, that generalizes architectures, datasets, and downstream methods, such as pruning or fine-tuning. The experiments are not intended to provide conclusive descriptions of phase transitions with the expectations of generality. Each of these merits its own investigation, and in fact, some of the effects have already been investigated in previous work. For example, phases in model pruning have already been established and have even been used to predict the optimal model for pruning [2].
>
> We believe the prevalence of phases in different applications makes phase transitions a powerful concept that should be considered - hence our dataset. Further, the loss-landscape metrics as established in previous work do explain some phase transitions, but fail to explain others. We see that not as a shortcoming of the metrics or the dataset, but as an excellent reason for further research to find better descriptors for these particular phase transitions. Ultimately, we believe this will build a deeper understanding of what makes models work for specific applications in specific conditions.

---

> > ### Author Response · Authors · 2024-11-22
> > **Author Response to Reviewer TH1T (continued)**
> >
> > ## Questions
> > - **CKA:** CKA is a correlation metric that is invariant to permutations, introduced in [3]. A higher CKA score indicates higher representation similarity.
> > - **Hyperparamter tuning:** We appreciate the question. The challenge of creating these datasets is making all of these choices in a consistent way, that leads to interesting variation in the result. In particular, we identified experiment setups that result in models with coverage of the known phases, as well as yet interesting variation in so-far-undescribed phases of other applications. We see this as our primary contribution. To the specific question: we have evaluated tuning the learning rate, but found it not necessary, likely due to the LR scheduler. Changing the learning rate for each mode size would directly affect temperature and therefore make the interpretation of the variation more difficult.
> > - Line 256: indeed, this should be IV-A. Our apologies.
> > - **Fine-tuning vs. transfer-learning:** We use fine-tuning to adapt to a new distribution (e.g. CIFAR10-> STL10) and transfer learning to adapt to a new task (Tiny-Imagenet -> STL10). We will adapt the captions in the phase plots to reflect that.
> >
> > [1] Yang et al., Taxonomizing local versus global structure in neural network loss landscapes, NeurIPS 2021.
> > [2] Zhou et al. A Three-regime model of Network Pruning ICML 2023.
> > [3] Kornblith et al., Similarity of Neural Network Representations Revisited, ICML 2019.
> >
> >
> >
> > -​​---
> >
> >
> > Once again, we thank the reviewer for the feedback. We hope to have addressed the reviewer’s questions and alleviate the concerns. We look forward to any further questions and the discussion.

---

> > > ### Comment · Reviewer_TH1T · 2024-11-28
> > >
> > > Thanks for your thoughtful reply.
> > >
> > > First of all, I completely agree with the authors that datasets have an important place at ICLR and are valuable research directions. But I also agree with the authors that as a "dataset paper" it should (mostly) be assessed based on the usefulness and impact of said dataset. This is where I still have some concerns.
> > >
> > > You mention that "The experiments [in Section 4] are not intended to provide conclusive descriptions of phase transitions with the expectations of generality" and I agree with that. However, I feel this highlights a limitation: Investigating each of these phenomena in sufficient detail to provide meaningful conclusions, would be impossible with the given dataset alone and instead would require substantial additional data. Therefore, I have doubts about the usefulness of the presented dataset.
> > >
> > > Nevertheless, I have increased my score as I believe this better reflects the overall quality of the paper and dataset. However, I still believe that this paper can and should be strengthened before being published.

---

### Official Review · Reviewer_tzPV · 2024-11-07

**Soundness:** 3
**Presentation:** 4
**Contribution:** 3
**Rating:** 6
**Confidence:** 3

**Summary:**

This paper introduces Phase Transition Model Zoos, a comprehensive collection of neural networks trained across various datasets, architectures, and methods, including fine-tuning and transfer learning. The authors present phase diagrams based on established metrics across diverse settings and include detailed experimental procedures for generating these model zoos.

**Strengths:**

- The paper provides a clear and accessible background, making it understandable for readers who may be unfamiliar with the topic.
- This paper offers an extensive benchmark of model zoos for studying phase transitions in neural networks, covering a variety of network architectures, datasets, and training methods, which is valuable for further research and analysis in this area.

**Weaknesses:**

- The number of runs (i.e., three) used to average the results may be insufficient to substantiate certain claims and could impact the robustness of the findings.
- As noted by the authors, these model zoos are restricted to classification models within the computer vision domain, which may limit generalizability.
- The learning rate scheduler likely influences the phase diagrams of neural networks [1], yet the model zoos do not include datasets trained without a learning rate scheduler, which could provide additional insights into phase behavior.

[1] Theisen, Ryan, et al. "When are ensembles really effective?." Advances in Neural Information Processing Systems 36 (2024).

**Questions:**

### Major
- While these model zoos offer valuable resources for exploring phase transitions in neural network training for both ML and statistical physics communities, their large size may present challenges for practical use. Can users download or access only specific parts of the zoos they need?

### Minor
- There is an inconsistency in the seed settings: lines 242 and 665 list seeds as {1, 2, 3}, while Table 1 uses {0, 1, 2}. Please ensure consistency in these settings.

---

> ### Author Response · Authors · 2024-11-22
> **Author Response to Reviewer tzPV**
>
> We would like to thank the reviewer for the constructive feedback. We have responded to questions regarding the scope and applications of the dataset in our “Response to All Reviewers”. We address the remaining questions individually below.
>
> > The number of runs [...] could impact the robustness of the findings.
>
> We appreciate the feedback and agree with the reviewer. Our results do seem to indicate some fluctuations and outliers. Considering our limitations in the computational budget, this too represents the inevitable tradeoff between breadth and depth. Nonetheless, we believe the dataset provides an excellent starting point for research. Should phase transitions fall in an area with larger uncertainty, researchers using our dataset can easily re-train specific combinations, or interpolate between existing nodes to locally increase the resolution.
>
> > The learning rate scheduler likely influences the phase diagrams.
>
> We agree with the reviewer. While not the focus of the submission, we touch upon the effect of LR scheduler / decaying in Section 3.2.
> Building on previous work, we use the notion of phases as qualitatively different behavior. From that point of view, it is less important how a specific phase was reached. With that in mind, we wanted to generate the populations with a training setup close to real-world setups, but also achieve good coverage of all phases. Interestingly, LR decay appears to increase phase IV - which may explain why LR decay is so effective. Early stopping proved effective in balancing the coverage of the different phases.
>
> > Accessibility of individual models.
>
> We are planning to publish a python interface along with the dataset, that allows specific slicing.
>
> > Inconsistencies in seed settings.
>
> Our apologies for the inconsistencies. We will clarify that in the draft. Ultimately, the code to reproduce the experiments will be published as ground truth.

---

> > ### Comment · Reviewer_tzPV · 2024-11-26
> >
> > I thank the authors for their detailed responses to my earlier questions.
> >
> > After reviewing the other reviewers' concerns and the authors' responses, I have decided to lower my score from 8 to 6.
> >
> > While I appreciate the paper's valuable contribution in providing a robust framework and benchmark of model zoos for studying phase transitions in neural networks, I share the concern that the practical applications of this work are not clearly substantiated.
> >
> > Given this limitation, I find it difficult to strongly recommend this work for acceptance at the conference.

---

### Author Response · Authors · 2024-11-22
**Response to all Reviewers**

We would like to thank the reviewers for their feedback. The dataset received unanimously positive feedback. The reviewers find “[p]hase transitions [...] an interesting lens to study neural networks” (TH1T), and judge the dataset an “extensive benchmark of model zoos for studying phase transitions in neural networks, covering a variety of network architectures, datasets, and training methods, which is valuable for further research and analysis in this area” (tzPV), the “comprehensive dataset is a great building block to accelerate research in this area” (TH1T), which “demonstrates the existence of phase transitions in a more diverse set of models/datasets compared to prior work” (JhyP) and consequently “highly appreciate the authors work for building a large scale of model zoos [...] to those who are interested in the foundational understanding of NNs”(Qvjj). We thank the reviewers for the recognition of both the research potential of phase transitions, as well as the value of our dataset.

However, the reviewers have raised criticism around the contribution, novelty, and application of our submission that we want to address here. We structure the following around the primary area of the submission, the scope of the dataset, and use cases.

## Primary Area: Dataset and Benchmark.

Several reviewers criticized the lack of novelty, methodological contributions, or new insights.
Among various topics, the ICLR call for papers also lists ‘datasets and benchmarks’. Datasets and benchmarks have previously been published at ICLR, e.g. [1,2,3,4,5].
We want to clarify that this submission is a dataset/benchmark paper, not a methods paper, as specified by the primary area of the submission.
As such, our primary contribution lies in providing a structured, high-quality dataset that enables systematic, reproducible research on phase transitions across neural networks.
As a dataset contribution, we believe it should be assessed based on its utility, value to the community, and the research that can be done with it, rather than on novel methods or theoretical insights. We are encouraged that the reviewers all appear to agree that our dataset offers substantial value as a resource for future research on phase transitions.
We hope that the reviewers will consider adjusting their scores accordingly.

## Scope of the Dataset
Several reviewers raised questions about the effect of breadth over depth of the dataset and asked for more tasks, larger datasets, larger models, and higher resolution grids.
We agree with the reviewers that extending the dataset that way is desirable and would improve the dataset.
However, we also want to highlight that our computational resources are not unlimited. That inherently leads to trade-offs between the breadth of datasets and the depth of each individual grid. We made the conscious decision to generalize the concept of phase transition across different datasets, architectures, and model sizes. We feel this provides better empirical support and allows research on translating phase transitions between the different scenarios.
We believe that our dataset is a necessary, first step for future research on phase transition and model zoos. As is, it already provides a valuable resource for applications, some of which we discuss in the next section.
Further, we invite contributions from the community to extend or refine the dataset where necessary further to improve the dataset and its scope of research applications.

---

> ### Author Response · Authors · 2024-11-22
> **Response to all Reviewer (continued)**
>
> ## Use Cases for the Dataset
> Reviewers have voiced questions about the applications for the dataset. As that is essential to a dataset contribution, we here summarize where we see the three main applications of the dataset. We want to stress that the dataset is not limited to it, and we invite the community to use it and adapt it to their needs. We will revise the submission to make these uses clearer.
>
> ### Research on Phase Transitions
> As the reviewers acknowledge, phase transitions are an interesting angle to understand the fundamental behavior of neural networks.
>
> The reviews raise excellent research questions that - while out of scope for a dataset paper - all merit answering. We discuss some related work in Section 2 of the submission. Examples for such research are the description of phases or identification of smoothness / sharpness of the phase transitions, as well as isolation of ‘order’ variables with phase transitions. In section 4, we demonstrate some of such order variables in application examples, to illustrate the prevalence of phases. Further, the existing loss landscape metrics seem to predict certain phase transitions in general model performance but do not seem to be sensitive to others. Future work may search for metrics to predict other phase transitions.
>
>
> We encourage the community to engage in that research to improve our understanding.
> That type of research requires datasets of trained models that systematically cover a variety of phases, and allow for validation of findings across different architectures and datasets. We therefore consider our dataset a basis for future research on phase transitions.
>
> ### Benchmarking Methods for Phase Aware Evaluation
> A key insight from the notion of phases is that point estimates for method evaluation are brittle and carry little information. We therefore call for researchers to explicitly evaluate their methods for phase transitions, to understand where their method works, and where its limits are. This not only helps evaluate the robustness of methods but also provides leads for why methods may or may not work that can inspire further research. Many methods explicitly use pre-trained models. Here, the performance of the eventual outcome directly depends on the pre-trained model. We present our dataset as a resource to systematically evaluate the impact of the phase of the pre-trained model on the outcome of the method. Illustrative examples for that are presented in Section 4 of the submission. We would like to emphasize that these examples are not meant to identify the specific phase transitions or their relation to loss landscape metrics, merely to illustrate the existence of different phases in different applications.
> Here, specifically, knowledge of what traits to look for in pre-trained models can not only lead to more reliable results but also build understanding.
>
> ### Population-Based Methods
> There is an entire research field that uses trained models as data modality. We give examples of methods in that field in Section 5 of the submission. Most work in that domain uses not individual models, but populations or zoos. As in general machine learning, the quality of learned models heavily depends on the quality of the dataset. Existing model zoo datasets often do not control for variation and just use some trained models, or vary some hyperparameters to achieve some variation in performance metrics, e.g. [6,7]. Instead, for our dataset, we make sure to cover the known phases, and therefore the range of model behavior on this task. In other words, we ensure the dataset contains the different flavors, rather than hoping they are covered.
>
>
> [1] Triantafillou et al., Meta-Dataset: A Dataset of Datasets for Learning to Learn from Few Examples, ICLR 2020.
> [2] Tian et al., A Control-Centric Benchmark for Video Predictions, ICLR 2023.
> [3] Li et al., A Minimalist Dataset for Systematic Generalization of Perception, Syntax, and Semantics, ICLR 2023.
> [4] Jung et al., Neural Architecture Design and Robustness: A Dataset, ICLR 2023.
> [5] Standley et al., An Extensible Multi-modal Multi-task Object Dataset with Materials, ICLR 2023.
> [6] Schuerholt et al., Model Zoo: A Dataset of Diverse Populations of Neural Network Models, NeurIPS 2022.
> [7] Unterthiner et al., Predicting neural network accuracy from weights, 2020.

---

### Meta-Review · Area_Chair_cojy · 2024-12-18

**Metareview:**

The authors propose a new dataset for studying phase transitions in neural networks trained. In contrast to prior work, this paper includes more datasets, models, and methods. Overall, it addresses an important phenomenon in Deep Learning and offers a chance to deeply study the optimization dynamics of large models.

However, in its current form, the paper has a series of shortcomings. The experiments that generated the data are based exclusively on computer vision models, bypassing the important domain of Large Language Models. Since training LLMs is where most of the computing resources are spent, it is natural to ask whether the findings on computer vision data also apply to NLP.

Furthermore, certain design choices are questionable in terms of the ability to deduce significant results. The dataset is based on averages of only three optimization runs. Last but not least, while the concept of having a dataset paper is not in itself a reason for rejection, the AC agrees with the reviewers that a deeper analysis is necessary to improve the novelty of the work, including a discussion of practical applications on how can this dataset lead to a concrete impact beyond a general understanding of the phase transitions.

As a result, I recommend the rejection of the paper and advise the reviewers to incorporate the comments into the next submission.

**Additional Comments On Reviewer Discussion:**

The authors engaged with the reviewers during the rebuttal, but the question by reviewer JhyP on the concrete application domains for the dataset remained largely unaddressed.

Furthermore, concerning the question of reviewer tzPV on the limited number of three runs per experiment, the authors did not provide new compelling evidence but replied by stating a necessary trade-off between computational resources and uncertainty.

---

### Decision · Program_Chairs · 2025-01-22

Reject